Original research

# Multimorbidity patterns in low-middle and high income regions: a multiregion latent class analysis using ATHLOS harmonised cohorts

Ivet Bayes-Marin [1,2,3] Albert Sanchez-Niubo [1,2] Laia Egea-Cortés [4]
Hai Nguyen [5] Matthew Prina [5] Daniel Fernández [2,6]
Josep Maria Haro [1,2,3] Beatriz Olaya [1,2]

For numbered affiliations see end of article.

**Correspondence to**
Dr Albert Sanchez-Niubo;
albert.sanchez@pssjd.org

## ABSTRACT

**Objectives** Our aim was to determine clusters of non-communicable diseases (NCDs) in a very large, population-based sample of middle-aged and older adults from low- and middle-income (LMICs) and high-income (HICs) regions. Additionally, we explored the associations with several covariates.

**Design** The total sample was 72140 people aged 50+ years from three population-based studies (English Longitudinal Study of Ageing, Survey of Health, Ageing and Retirement in Europe Study and Study on Global Ageing and Adult Health) included in the Ageing Trajectories of Health: Longitudinal Opportunities and Synergies (ATHLOS) project and representing eight regions with LMICs and HICs. Variables were previously harmonised using an ex-post strategy. Eight NCDs were used in latent class analysis. Multinomial models were made to calculate associations with covariates. All the analyses were stratified by age (50–64 and 65+ years old).

**Results** Three clusters were identified: 'cardio-metabolic' (8.93% in participants aged 50–64 years and 27.22% in those aged 65+ years), 'respiratory-mental-articular' (3.91% and 5.27%) and 'healthy' (87.16% and 67.51%). In the younger group, Russia presented the highest prevalence of the 'cardio-metabolic' group (18.8%) and England the 'respiratory-mental-articular' (5.1%). In the older group, Russia had the highest proportion of both classes (48.3% and 9%). Both the younger and older African participants presented the highest proportion of the 'healthy' class. Older age, being woman, widowed and with low levels of education and income were related to an increased risk of multimorbidity. Physical activity was a protective factor in both age groups and smoking a risk factor for the 'respiratory-mental-articular'.

**Conclusion** Multimorbidity is common worldwide, especially in HICs and Russia. Health policies in each country addressing coordination and support are needed to face the complexity of a pattern of growing multimorbidity.

## BACKGROUND

By 2050, the population aged 60 years and older is expected to reach 2 billion worldwide compared with 900 million in 2015.[1] Along with this rapid increase, the

---

**Strengths and limitations of this study**

► This study used a large, harmonised, multiregional database, which allowed us to compare two age groups as well as disease prevalence in regions with differing incomes.

► The presence or absence of the non-communicable diseases was based on self-reported measures, and thus might be affected by measurement errors or lack of accuracy.

► Only common diseases across studies were included in the analyses, so this might have led to a smaller number of latent classes or to different clusters.

► When performing latent class analysis, we forced the solution as we aimed to do comparisons among age subsamples and regions in terms of disease prevalence as well as protective and risk factors.

► The use of multiple imputations for missing data in the covariates could carry some bias.

---

incidence of chronic conditions (CCs) or non-communicable diseases (NCDs) is also on the rise, having become the leading cause of morbidity and disability worldwide.[2]

Multimorbidity, defined as the co-existence of two or more CCs, is more common in older adults and is often more prevalent in people of lower socioeconomic status.[3] Multimorbidity is thought to account for 65% of total healthcare expenses in high-income countries (HICs) because of the huge associated healthcare utilisation.[4] Due to the increasing prevalence of multimorbidity, the managing of multiple conditions has become an unavoidable international research priority because of the high impact on the quality of life of patients and caregivers and on healthcare systems.[3]

Most studies on the prevalence of multimorbidity in older people come from HICs, while data from middle-aged adults and

low- and middle-income countries (LMICs) are much more limited.[5–8] LMICs are experiencing an increase in life expectancy that, together with changes in lifestyle and environment exposures, is triggering changes in their disease burden profile.[3 9] Few studies have compared patterns of multimorbidity between HICs and LMICs. Afshar *et al*[10] used population-based chronic disease data from the World Health Survey to compare multimorbidity prevalence across 27 LMICs and 1 HIC and used gross domestic product (GDP) to study intercountry socio-economic differences. They found high multimorbidity prevalence in all countries and a positive but non-linear relationship between country GDP and multimorbidity prevalence, suggesting the influence of other factors, such as lifestyles, social conditions and differences across health systems. Four latent classes were identified in a cross-sectional sample of Australian seniors aged 50 years and over, using self-reported diagnosis of 11 conditions, including cancer and Parkinson's disease.[11] Another study, focusing on complex healthcare needs of Italian elderly people, found five clusters using 15 diseases.[12] A study conducted in a sample of 162 283 people from a survey of Danish population identified seven latent classes considering 15 chronic diseases and seven age groups, ranging from 16 to 104 years.[13] These differences could be explained in light of variations in collection methods, data sources, populations, diseases included and the analysis performed.[11 14 15]

Similarly, the lack of study of differences in multimorbidity between HICs and LMICs may be due to the use of different methodologies, which might hinder comparisons of prevalence and multimorbidity patterns across countries. The integration of data from different studies would allow us to determine differences across regions and cohorts, as well as to explore risk and protective factors involved in the clustering of CCs, thereby improving our understanding of the problem and the creation of adapted medical guidelines.

This study aimed to (a) identify multimorbidity clusters in middle-aged (50–64 years) and older adults (+65 years) from different regions, classified as LMICs and HICs; (b) investigate the associations between multimorbidity clusters and sociodemographic, economic, lifestyles and health status variables and (c) explore differences across regions.

## METHODS
### Study design and data extraction
The present study used data from the Ageing Trajectories of Health: Longitudinal Opportunities and Synergies (ATHLOS) project.[16] Longitudinal data from 17 international cohort studies related to health and ageing were harmonised with the aim of obtaining an integrated dataset and achieving a better understanding of ageing and health processes.

We selected three studies due to their inclusion of the variables of interest and the possibility of comparing HICs and LMICs. Baseline samples of the following studies were included in the analyses: the WHO's Study on Global Ageing and Adult Health (SAGE),[17] the English Longitudinal Study of Ageing (ELSA)[18] and the Survey of Health, Ageing and Retirement in Europe Study (SHARE).[19] These panel studies included non-institutionalised people aged 50 years and older. SAGE comprises six LMICs according to The World Bank Classification,[20] namely Ghana, South Africa, Mexico, India, China and Russia; ELSA includes the English population and SHARE covers 11 countries of the European Union and Israel at baseline, considered as HICs.[20]

The analyses presented focused on people aged 50 years and older who were part of the core sample of each study and who completed a non-proxy interview at baseline. We excluded from the analyses those participants who participated via proxy due to cognitive problems or severe physical limitations. Moreover, people with missing values in sex and age were excluded, resulting in a final sample of 72 140 individuals. Mexico was excluded from the analyses due the high percentage of missingness in the variables of interest (see online additional file 1: table S1).

### Patient and public involvement
No patient involved.

### Variables
The following variables were the result of a stringent, ex-post harmonisation process using systematic harmonisation methodology and tools from Maelstrom Research.[21]

Eight NCDs were used to conduct the analysis, including those that were available in the three studies: diabetes, hypertension, asthma, chronic lung disease, joint disorders (arthritis, rheumatism or osteoarthritis), angina or myocardial infarction, stroke and depression. The presence or absence of these conditions was self-reported and based on a medical diagnosis. Depression was assessed with standardised tools, such as the Composite International Diagnostic Interview (CIDI) in the SAGE study, the Center for Epidemiologic Studies Depression Scale (CES-D) in ELSA and the EURO-D in SHARE.[22–24] A dichotomous variable (yes/no) was created using the indicated cut-off score for each tool and population based on previous studies.[22 25 26]

Self-reported demographic variables included age, sex, level of education (primary or less, secondary and tertiary), marital status (single, married or currently cohabiting, separated or divorced, and widowed) and quintiles of household wealth (first quintile indicating lowest level). Lifestyles and health behaviours were 'ever smoked' any type of tobacco and physical activity referring to the practice of vigorous exercise during the last 2 weeks, both coded as *yes* or *no*. Other health-related variables were self-rated health (good, moderate or poor), presence or absence of loneliness feelings in the last week, difficulties in activities of daily living (ADL), cognitive performance and number of diseases.

To assess difficulties in ADL, we used a set of daily self-care activities, which were common across studies, such as problems in using the toilet, bathing or showering, getting dressed, eating, moving, or getting in or out of bed. Each of the ADL difficulties was coded into a *yes*/*no* if the person answered 'severe' or 'extreme/cannot do it'. To build the set of ADL difficulties, we coded *yes* if the person reported at least one difficulty in any of the six items.

Immediate and delayed recall was assessed using the 10-word learning list task and verbal fluency utilising the animal naming test.[27] Continuous total scores were used to perform the analyses. Number of diseases was built by adding up the occurrences of all the above-mentioned NCDs.

Finally, a seven-level regional membership variable was created in order to analyse regional differences, based on the WHO and the United Nations Statistical Division (UNSD) regional classification.[28 29] Moreover, the World Bank Classification was used to classify these regions into HICs or LMICs.[20] SAGE includes Africa (Ghana and South Africa), China and India, all of them considered as LMICs. SHARE countries were grouped into three regions: Northern Europe (Denmark, Sweden), Southern Europe (Greece, Italy and Spain) and Western Europe (Austria, Belgium, France, Germany, Israel, Netherlands and Switzerland). ELSA and SHARE regions were considered as HICs. Ghana and South Africa were grouped together and named as Africa for practical purposes as well as due to their smaller sample size. These countries are not necessarily representative of the whole continent.

## Statistical analysis

All the analyses were performed using data from the baseline. Descriptive statistics were used to summarise information regarding sociodemographic economic variables and disease prevalence among regions. CIs (95% CI) were calculated for categorical variables in order to make comparisons across regions.

Latent class analysis (LCA) was conducted stratified by age (50–64, +65 years). Eight NCDs (diabetes, hypertension, asthma, chronic lung disease, joint disorders, angina–myocardial infarction, stroke and depression) were used as observed indicators without using covariates since we aimed to identify latent classes only based on disease variables. Region was used as cluster when conducting LCA in order to accurately describe disease proportions, indicating that the subjects were not independent random draws, but rather were nested within clusters.[30]

The optimal number of latent classes was determined using the adjusted Bayesian Information Criterion (aBIC), the consistent Akaike Information Criterion (CAIC) and the Entropy Index. Lower values of aBIC and CAIC indicate better fit, whereas Entropy Index values higher than 0.80 indicate that the latent classes are highly discriminating.[31] The average posterior probability indicates how well a model classifies individuals into their most likely class. Values higher than 0.70 indicate well-identified classes.[32] Additionally, interpretability and clinical judgement were used.[32 33]

Missing data in one of the indicators were handled with the full information maximum likelihood technique, assuming missing-at-random (MAR).[34] Missing data in the covariates were handled using multiple imputation by chained equations assuming MAR.[34] The imputation model included the outcome (group membership in one of the latent classes) and all the variables used in the regression models. In the online additional file 1: table S2-10, there is a report of those variables and the percentage of missingness of each region in the variables of interest.

Adjusted multinomial logistic regression models were used to assess the association between the outcome (multimorbidity classes, with the 'healthy' class as the reference category) and several variables: loneliness, ever smoked, physical activity, limitations in ADL, self-rated health, immediate recall, delayed recall and verbal fluency. The model was additionally adjusted for sex, age, marital status, education level, wealth and the region at baseline. Due to potential collinearity between income and education, we checked the significance and magnitude of the correlation between both variables. The association was small, and thus, both covariates were included as separate variables in the models. Regression models were conducted separately in 100 imputed datasets and results combined using Rubin's rules.[35]

All analyses were conducted with Stata SE V.13.1. LCAs were performed using a Stata plugin.[30]

## RESULTS
### Descriptive analysis

In table 1, the main characteristics of the sample by region are presented. The mean age ranged from 62 years (SD=9.02) in Southern Asia to 65 years (10.18) in Russia and 65 years (10.26) in England. Some 54% were women, 72% were married or cohabitating, and 39% had secondary education. Russia presented the highest number of conditions (mean 1.66) compared with Africa (0.64), China (0.80) and India (0.72).

The most prevalent conditions in the total sample were hypertension (31.2%, 95% CI=30.9% to 31.6%) and joint disorders (22.4%, 95% CI=22.0% to 22.7%). Hypertension was particularly high in Russia (56.5%, 95% CI=54.9% to 58.1%) compared with the other regions. Diabetes prevalence was greater in Southern (11.9%, 95% CI=11.2% to 12.7%) and Western Europe (10.4%, 95% CI=9.9% to 10.9%), whereas Africa and China presented the lowest proportions. Similarly, myocardial infarction–angina was highly prevalent in Russia (33.1%, 95% CI=31.6% to 34.6%), followed by countries of Northern (13.8%, 95% CI=12.8% to 14.8%), Southern (11.7%, 95% CI=11.0% to 12.4%) and Western Europe (13.1%, 95% CI=12.6% to 13.6%).

**Table 1** Main characteristics of the total sample and by regions

| Region | Total N=72140 | Africa* n=7950 | China† n=12840 | India† n=6558 | Russia† n=3887 | England‡ n=11 517 | Northern Europe§ n=4573 | Southern Europe¶ n=7465 | Western Europe** n=17 350 |
|---|---|---|---|---|---|---|---|---|---|
| Age, mean (SD) | 64.05 (9.96) | 63.60 (10.26) | 63.07 (9.31) | 61.80 (9.02) | 65.06 (10.18) | 65.06 (10.26) | 64.78 (10.34) | 65.02 (10.17) | 64.34 (9.96) |
| Woman, % (95% CI) | 54.0 (53.6 to 54.4) | 52.3 (51.2 to 53.4) | 52.9 (52.1 to 54.8) | 49.6 (48.4 to 50.8) | 64.6 (63.0 to 66.1) | 54.6 (53.7 to 55.5) | 53.2 (51.8 to 54.7) | 55.4 (54.2 to 56.5) | 54.1 (53.4 to 54.9) |
| **Marital status, % (95% CI)** | | | | | | | | | |
| Single | 4.1 (4.0 to 4.3) | 6.7 (6.2 to 7.3) | 0.9 (0.7 to 1.2) | 1.0 (0.8 to 1.2) | 2.8 (2.3 to 3.4) | 5.0 (4.6 to 5.4) | 5.5 (4.8 to 6.2) | 6.3 (5.8 to 6.9) | 4.9 (4.6 to 5.3) |
| Married | 71.5 (71.1 to 72.0) | 54.5 (53.4 to 55.6) | 83.5 (82.8 to 84.1) | 74.1 (73.1 to 75.2) | 56.1 (54.5 to 57.7) | 69.1 (68.2 to 69.9) | 72.4 (71.0 to 73.7) | 73.3 (72.3 to 74.3) | 73.3 (72.7 to 74.0) |
| Divorced | 5.9 (5.7 to 6.2) | 10.4 (9.7 to 11.1) | 1.7 (1.5 to 2.0) | 0.6 (0.5 to 0.9) | 8.3 (7.4 to 9.2) | 9.0 (8.5 to 9.5) | 9.6 (8.8 to 10.5) | 2.6 (2.3 to 3.0) | 6.8 (6.5 to 7.2) |
| Widowed | 18.4 (18.1 to 18.6) | 27.3 (26.3 to 28.3) | 13.9 (13.3 to 14.6) | 24.3 (23.2 to 25.3) | 32.7 (31.2 to 34.2) | 16.9 (16.3 to 17.6) | 12.5 (11.6 to 13.5) | 17.8 (16.9 to 18.6) | 14.9 (14.4 to 15.4) |
| **Education level % (95% CI)** | | | | | | | | | |
| Primary or less | 33.4 (33.0 to 33.7) | 30.5 (29.5 to 31.5) | 37.9 (37.1 to 38.8) | 25.5 (24.5 to 26.6) | 9.9 (9.0 to 10.9) | 42.4 (41.5 to 43.4) | 29.6 (28.3 to 30.9) | 59.4 (58.3 to 60.5) | 23.2 (22.6 to 23.8) |
| Secondary | 38.7 (38.4 to 39.1) | 19.0 (18.2 to 19.9) | 33.2 (32.4 to 34.0) | 18.2 (17.3 to 19.2) | 69.3 (67.9 to 70.8) | 37.5 (36.7 to 38.4) | 45.1 (43.6 to 46.5) | 31.3 (30.2 to 32.3) | 55.1 (54.3 to 55.8) |
| Tertiary | 12.0 (11.9 to 12.2) | 3.9 (3.5 to 4.3) | 4.7 (4.3 to 5.0) | 5.0 (4.4 to 5.5) | 19.7 (18.5 to 21.0) | 11.1 (10.6 to 11.7) | 24.2 (23.0 to 25.5) | 9.1 (8.4 to 9.7) | 20.8 (20.2 to 21.4) |
| **Wealth, quintiles % (95% CI)** | | | | | | | | | |
| First (worse) | 19.3 (19.1 to 19.6) | 19.3 (18.4 to 20.2) | 19.8 (19.1 to 20.5) | 16.2 (15.3 to 17.1) | 18.2 (17.0 to 19.5) | 19.0 (18.3 to 19.7) | 20.7 (19.6 to 21.9) | 20.3 (19.4 to 21.2) | 19.9 (19.3 to 20.5) |
| Second | 19.7 (19.4 to 20.0) | 19.7 (18.4 to 20.2) | 19.7 (19.0 to 20.4) | 18.6 (17.6 to 19.5) | 19.8 (18.5 to 21.1) | 19.3 (18.6 to 20.1) | 20.4 (19.3 to 21.6) | 20.4 (19.5 to 21.3) | 19.9 (19.4 to 20.6) |
| Third | 19.7 (19.4 to 19.9) | 19.8 (18.9 to 20.7) | 20.1 (19.4 to 20.8) | 18.4 (17.5 to 19.4) | 20.3 (19.0 to 21.6) | 19.7 (18.9 to 20.4) | 20.1 (19.0 to 21.3) | 19.8 (18.9 to 20.7) | 19.5 (18.9 to 20.1) |
| Fourth | 19.9 (19.7 to 20.3) | 20.5 (19.6 to 21.4) | 20.5 (19.8 to 21.2) | 21.5 (20.5 to 22.5) | 20.0 (18.8 to 21.3) | 19.6 (18.9 to 20.3) | 19.7 (18.5 to 20.8) | 19.4 (18.6 to 20.4) | 19.3 (18.8 to 19.9) |
| Fifth (best) | 20.1 (19.8 to 20.4) | 20.4 (19.5 to 21.3) | 19.7 (19.0 to 20.4) | 24.8 (23.8 to 25.9) | 21.6 (20.3 to 22.9) | 19.6 (18.9 to 20.3) | 18.9 (17.8 to 20.1) | 18.9 (18.0 to 19.8) | 19.4 (18.8 to 20.0) |
| No diseases, mean (SD) | 1.02 (1.14) | 0.64 (0.94) | 0.80 (0.99) | 0.72 (0.97) | 1.66 (1.38) | 1.19 (1.13) | 1.02 (1.10) | 1.28 (1.25) | 1.10 (1.16) |
| **Diseases, % (95% CI)** | | | | | | | | | |
| Diabetes | 8.5 (8.3 to 8.7) | 6.6 (6.1 to 7.2) | 6.5 (6.1 to 7.0) | 7.3 (6.7 to 7.9) | 9.0 (8.1 to 10.0) | 7.4 (6.9 to 7.9) | 8.2 (7.4 to 9.0) | 11.9 (11.2 to 12.7) | 10.4 (9.9 to 10.9) |
| Hypertension | 31.2 (30.9 to 31.6) | 21.5 (20.6 to 22.4) | 27.4 (26.6 to 28.2) | 17.5 (16.6 to 18.5) | 56.5 (54.9 to 58.1) | 37.8 (36.9 to 38.7) | 29.3 (28.0 to 30.7) | 35.6 (34.5 to 36.7) | 32.3 (31.6 to 33.0) |
| Joint disorders | 22.4 (22.0 to 22.7) | 17.8 (16.9 to 18.6) | 22.1 (21.4 to 22.8) | 17.9 (17.0 to 18.9) | 35.2 (33.7 to 36.7) | 32.5 (31.6 to 33.3) | 15.7 (14.6 to 16.8) | 26.3 (25.3 to 27.3) | 16.8 (16.3 to 17.4) |

Continued

**Table 1** Continued

| Region | Total N=72140 | Africa* n=7950 | China† n=12840 | India† n=6558 | Russia† n=3887 | England‡ n=11 517 | Northern Europe§ n=4573 | Southern Europe¶ n=7465 | Western Europe** n=17 350 |
|---|---|---|---|---|---|---|---|---|---|
| Asthma | 5.5 (5.3 to 5.6) | 4.1 (3.7 to 4.5) | 2.4 (2.2 to 2.7) | 6.9 (6.3 to 7.6) | 3.4 (2.9 to 4.0) | 11.7 (11.1 to 12.3) | 7.6 (6.8 to 8.4) | 4.1 (3.7 to 4.6) | 4.1 (3.8 to 4.4) |
| Chronic lung disease | 6.1 (5.9 to 6.3) | 1.5 (1.2 to 1.7) | 8.6 (8.1 to 9.1) | 4.1 (3.6 to 4.6) | 17.9 (16.8 to 19.2) | 6.5 (6.1 to 7.0) | 4.5 (3.9 to 5.2) | 5.4 (4.9 to 6.0) | 4.9 (4.6 to 5.3) |
| MI–angina | 10.0 (9.8 to 10.3) | 4.5 (4.1 to 5.0) | 8.8 (8.3 to 9.3) | 4.9 (4.4 to 5.5) | 33.1 (31.6 to 34.6) | 3.3 (3.0 to 3.6) | 13.8 (12.8 to 14.8) | 11.7 (11.0 to 12.4) | 13.1 (12.6 to 13.6) |
| Stroke | 3.8 (3.7 to 3.9) | 3.2 (2.8 to 3.6) | 3.5 (3.2 to 3.8) | 2.2 (1.9 to 2.6) | 6.0 (5.3 to 6.8) | 4.5 (4.1 to 4.9) | 5.0 (4.4 to 5.7) | 3.1 (2.8 to 3.6) | 3.9 (3.7 to 4.2) |
| Depression | 15.3 (15.0 to 15.5) | 5.8 (5.3 to 6.3) | 1.2 (1.1 to 1.4) | 12.1 (11.3 to 12.9) | 5.2 (4.5 to 5.9) | 16.5 (15.8 to 17.2) | 18.8 (17.7 to 20.0) | 31.7 (30.6 to 32.7) | 25.0 (24.3 to 25.6) |

The analyses were performed before multiple imputation procedure.
*SAGE study—Africa: Ghana, South Africa.
†SAGE study.
‡ELSA study—England.
§SHARE study—Northern Europe: Denmark, Sweden.
¶SHARE study—Southern Europe: Greece, Italy, Spain.
**SHARE study—Western Europe: Austria, Belgium, France, Germany, Israel, Netherlands, Switzerland.
ELSA, English Longitudinal Study of Ageing; SAGE, Study on Global Ageing and Adult Health; SHARE, Survey of Health, Ageing and Retirement in Europe Study.

Joint disorders were more prevalent in Russia (35.2%, 95% CI=33.7% to 36.7%) and England (32.5%, 95% CI=31.6% to 33.3%). The prevalence of asthma was greater in England than other regions (11.7%, 95% CI=11.1% to 12.3%) and chronic lung disease was greater in Russia (17.9%, 95% CI=16.8% to 19.2%).

As for the prevalence of depression, European countries presented the highest values, especially in Southern (31.7%, 95% CI=30.6% to 32.7%) and Western Europe (25.0%, 95% CI=24.3% to 25.6%), whereas LMICs showed very low proportions, especially in China, where only 1.2% of people aged 50+ years presented depression.

## Multimorbidity patterns

Table 2 displays the aBIC, CAIC and entropy values, proportions and average posterior probability of each latent class, for a two-class to five-class model in both age subsamples. In the younger subsample (50–64 years), the five-class solution yielded the lowest aBIC and CAIC values and the highest entropy value (0.67). However, it was dismissed because one of the latent classes was very infrequent and the posterior probabilities were far below 0.70. Similarly, the four-class model was rejected for an inadequate posterior probability value in one of the classes (0.52). The model finally selected was the three-class model. The three-class solution was also chosen for the older age group because of lower posterior probability values in the four-class and five-class models in spite of lower aBIC and CAIC values.

We named each latent class according to the most prevalent diseases within each latent class. Figure 1 shows the distribution of each condition across the three latent classes ('cardio-metabolic', 'respiratory-mental-articular' and 'healthy' class) in the total sample and by regions. The 'cardio-metabolic' class presented excess prevalence of diabetes, hypertension, myocardial infarction or angina and stroke, comprising 8.93% of the total sample in the younger group and 27.22% in the older group. The 'respiratory-mental-articular' class, which comprised 3.91% and 5.27% of each sample, respectively, showed greater prevalence of joint disorders, asthma, chronic lung diseases and depression. Finally, the 'healthy' class presented low prevalence of conditions, comprising 87.16% of the sample in the first age group and 67.51% in the second group.

Differences in the proportions of multimorbidity classes were found across regions (figure 1). The 'cardio-metabolic' class (18.8%, 95% CI=17.1% to 20.6%) was significantly greater in Russia than in other regions, and England (5.1%, 95% CI=4.5% to 5.7%) showed a higher proportion of individuals classified into the 'respiratory-mental-articular' class. The 'healthy' class was higher in Africa (91.5%, 95% CI=90.7% to 92.3%), China (90.8%, 95% CI=90.1% to 91.4%) and India (89.5%, 95% CI=88.5% to 90.4%), and remarkably lower in Russia (71.6%, 95% CI=69.5% to 73.6%) compared with other regions.

**Table 2** Comparison between models in individuals aged 50–64 and +65 years

| No of latent classes | Aged 50–64 years | | | | | Aged ≥65 years | | | | |
|---|---|---|---|---|---|---|---|---|---|---|
| | Information criteria indices | | Classification quality | Latent classes, n (%) | Average posterior probability | Information criteria indices | | Classification quality | Latent classes, n (%) | Average posterior probability |
| | aBIC | CAIC | Entropy | | | aBIC | CAIC | Entropy | | |
| 2 | 1512.91 | 1583.94 | 0.51 | 33 023 (82.15) | 0.88 | 1603.3 | 1674.33 | 0.39 | 21 113 (66.10) | 0.83 |
| | | | | 7177 (17.85) | 0.75 | | | | 10 827 (33.90) | 0.75 |
| **3** | **875.23** | **983.86** | **0.63** | **3589 (8.93)** | **0.76** | **1032.83** | **1141.46** | **0.5** | **8693 (27.22)** | **0.71** |
| | | | | **1571 (3.91)** | **0.67** | | | | **1684 (5.27)** | **0.68** |
| | | | | **35 040 (87.16)** | **0.87** | | | | **21 563 (67.51)** | **0.81** |
| 4 | 777.14 | 923.37 | 0.43 | 25 701 (63.93) | 0.52 | 817.6 | 963.83 | 0.63 | 9557 (29.92) | 0.65 |
| | | | | 626 (1.56) | 0.72 | | | | 1474 (4.61) | 0.78 |
| | | | | 7754 (19.29) | 0.8 | | | | 17 220 (53.91) | 0.86 |
| | | | | 6119 (15.22) | 0.68 | | | | 3689 (11.55) | 0.77 |
| 5 | 661.04 | 844.87 | 0.67 | 4578 (11.39) | 0.64 | 689.22 | 873.05 | 0.59 | 11 094 (34.73) | 0.77 |
| | | | | 247 (0.61) | 0.67 | | | | 1094 (3.43) | 0.71 |
| | | | | 32 423 (80.65) | 0.85 | | | | 14 155 (44.32) | 0.76 |
| | | | | 1359 (3.38) | 0.48 | | | | 1148 (3.59) | 0.63 |
| | | | | 1593 (3.96) | 0.64 | | | | 4449 (13.93) | 0.72 |

Boldface indicates the final selected model.

aBIC, adjusted Bayesian Information Criterion; CAIC, consistent Akaike Information Criterion.

Similar results were found for the older group (figure 2). In Russia, the 'cardio-metabolic' class was significantly higher than in other regions (48.3%, 95% CI=46.1% to 50.6%), whereas the 'healthy' class was the least frequent class compared with the rest of regions (38.4%, 95% CI=36.2% to 40.6%), followed by Southern Europe (52.6%, 95% CI=50.9 to 54.2). Africa and India showed lower proportions of individuals classified into the 'cardio-metabolic' class (12.9%, 95% CI=11.8% to 14.1% and 11.2%, 95% CI=10.0% to 12.6%, respectively).

### Association between multimorbidity classes and covariates

In the online additional file 2: table S1, the unadjusted relative risk ratios (RRRs) for both age subsamples are presented. The 'healthy' class was used as the reference group. In the case of the younger subsample, and compared with the 'healthy' class, individuals classified into the 'cardio-metabolic' and 'respiratory-mental-articular' classes were more likely to be older (RRR=1.09, 95% CI=1.08 to 1.10; RRR=1.06, 95% CI=1.04 to 1.07, respectively) and being widowed (RRR=1.4, 95% CI=1.1 to 1.7) and divorced in the 'respiratory-mental-articular' class (RRR=1.7, 95% CI=1.2 to 2.4). Being a man, having tertiary education and high levels of wealth had a protective effect for being in both multimorbidity groups compared with the 'healthy' class. Similarly, those individuals from the older subsample who were in the fourth and fifth quintile (RRR=0.8, 95% CI=0.7 to 1.0) were less likely to be classified into the 'respiratory-mental-articular' class compared with the 'healthy' group.

Regarding the association of regions and multimorbidity groups, some differences were found in the younger subsample. Taking Africa as the reference category, participants from Russia were more likely to be classified into the 'cardio-metabolic' class (RRR=3.6, 95% CI=3.0 to 4.2), whereas individuals from England (RRR=5.6, 95% CI=4.1 to 7.7), Northern Europe (RRR=2.8, 95% CI=1.9 to 4.1) and India (RRR=2.2, 95% CI=1.5 to 3.2) showed higher risk of being in the 'respiratory-mental-articular' class. In the case of the older subsample, all regions had greater risk of being classified into the 'respiratory-mental-articular' class compared with the 'healthy' class, especially participants from Russia (RRR=14.5, 95% CI=10.3 to 20.3) compared with Africa.

Table 3 shows the adjusted RRRs for both age subsamples, taking the 'healthy' class as the reference group. Both multimorbidity classes (cardio-metabolic and respiratory-mental-articular) were associated with all the covariates in the younger group, except for smoking status in the 'cardio-metabolic' class.

In the younger individuals subsample, both latent classes were more likely to be associated with the presence of feelings of loneliness (RRR=1.8, 95% CI=1.7 to 2.0; RRR=2.5, 95% CI=2.0 to 3.0), limitations in ADL (RRR=3.2, 95% CI=2.9 to 3.6; RRR=3.9, 95% CI=3.3 to 4.7) and worse health status (RRR=12.8, 95% CI=11.3 to 14.4; RRR=12.9, 95% CI=10.5 to 16.0). Physical activity had a protective effect for being in these classes and having smoked was a risk factor only for being classified into the 'respiratory-mental-articular' class (RRR=1.5, 95% CI=1.2 to 1.7). Conversely, those older individuals who had ever smoked had a higher risk of being in the 'respiratory-mental-articular' group (RRR=1.8, 95%

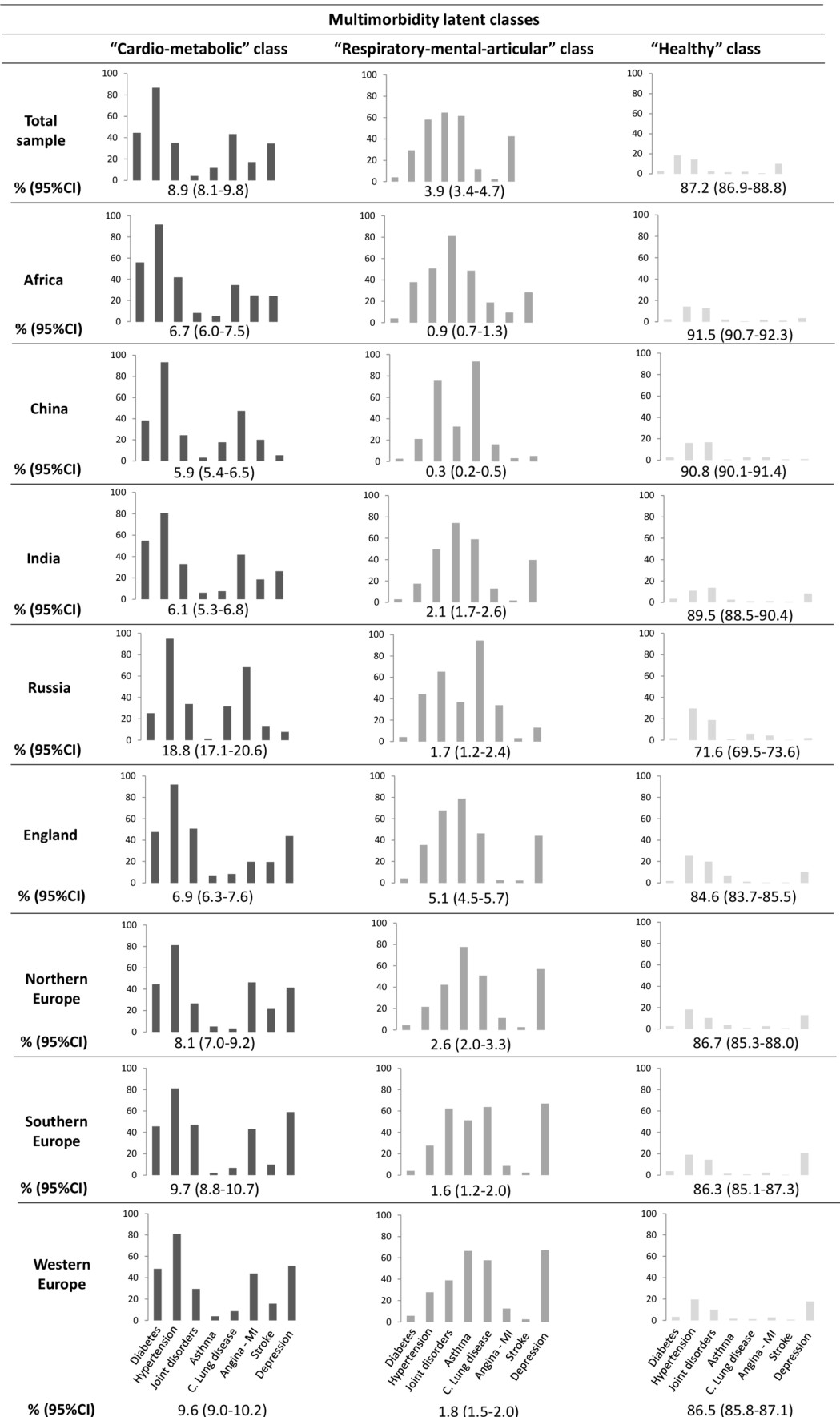

**Figure 1** Prevalence of diseases in the three latent classes in the total sample and by regions (subsample 50–64 years). C., chronic; MI, myocardial infarction.

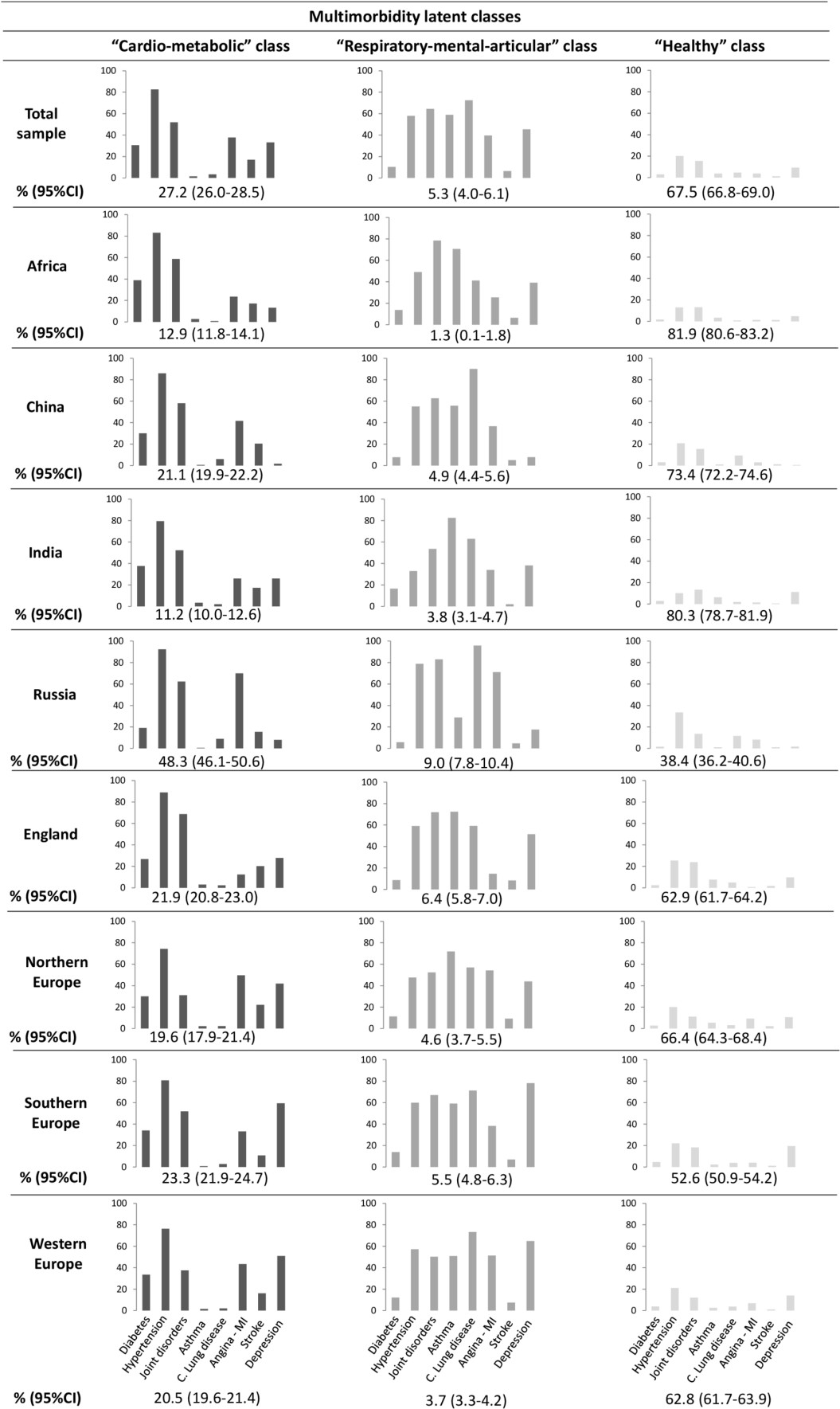

**Figure 2** Prevalence of diseases in the three latent classes in the total sample and by regions (subsample +65 years). C., chronic; MI, myocardial infarction

**Table 3** Association between latent multimorbidity membership and outcomes in individuals aged 50–64 and ≥65 years

| Variables* | Aged 50–64 years | | Aged ≥65 years | |
| --- | --- | --- | --- | --- |
| | 'Cardio-metabolic' class | 'Respiratory-mental-articular' class | 'Cardio-metabolic' class | 'Respiratory-mental-articular' class |
| Loneliness (yes/no) | 1.8 (1.7 to 2.0) | 2.5 (2.0 to 3.0) | 1.3 (1.2 to 1.5) | 1.9 (1.7 to 2.3) |
| Ever smoked (yes/no) | 1.1 (1.0 to 1.2) | 1.5 (1.2 to 1.7) | 1.0 (0.9 to 1.0) | 1.8 (1.5 to 2.0) |
| Physical activity (yes/no) | 0.4 (0.4 to 0.5) | 0.5 (0.5 to 0.6) | 0.5 (0.5 to 0.6) | 0.4 (0.4 to 0.5) |
| Limitations in ADL (yes/no) | 3.2 (2.9 to 3.6) | 3.9 (3.3 to 4.7) | 2.3 (2.1 to 2.5) | 4.0 (3.5 to 4.6) |
| Self-rated health | | | | |
| Good | 1 | 1 | 1 | 1 |
| Moderate | 4.8 (4.3 to 5.2) | 3.9 (3.2 to 4.7) | 3.1 (2.9 to 3.4) | 5.7 (4.8 to 6.8) |
| Poor | 12.8 (11.3 to 14.4) | 12.9 (10.5 to 16.0) | 6.2 (5.6 to 6.9) | 19.4 (16.2 to 23.4) |
| Memory: Immediate recall | 0.9 (0.9 to 0.9) | 0.9 (0.9 to 1.0) | 0.9 (0.9 to 0.9) | 0.9 (0.9 to 1.0) |
| Memory: Delayed recall | 0.9 (0.9 to 0.9) | 0.9 (0.9 to 1.0) | 0.9 (0.9 to 0.9) | 0.9 (0.9 to 1.0) |
| Verbal fluency | 1.0 (1.0 to 1.0) | 1.0 (1.0 to 1.0) | 0.9 (0.9 to 1.0) | 1.0 (1.0 to 1.0) |

The reference group for the multimorbidity group variable was the 'healthy' class.
Relative risk ratios (95% CI) from multinomial logistic regression models.
Models were run in 100 imputed datasets and results combined using Rubin's rules.
*Adjusted for sex, age, marital status, education level, wealth and region.
ADL, activities of daily living.

CI=1.5 to 2.0) and a lower risk of being classified into the 'cardio-metabolic' class (RRR=1.0, 95% CI=0.9 to 1.0). For both age subsamples, better performance in verbal memory was significantly associated with less risk of being classified into the two multimorbidity classes. Similarly, higher scores in verbal fluency were a protective factor for multimorbidity compared with the healthy individuals group.

## DISCUSSION
To the best of our knowledge, this is the first multiregion study to use harmonised data to compare multimorbidity patterns across different regions from three distinct population-based cohorts. We identified three latent classes of multimorbidity based on the presence of eight NCDs: the 'cardio-metabolic', the 'respiratory-mental-articular' and the 'healthy' class. The same clusters were identified in another study using SAGE original data, applying exploratory factor analysis in a sample of 41 909 individuals aged 50 years or older.[36] Similarly, a study of a representative sample of Spanish community-dwelling adults over 50 years old also found three latent classes using 11 CCs, showing similar disease distributions among the multimorbidity clusters.[37]

In our study, for both age groups the majority of the sample was classified into the 'healthy' class, 87.16% and 67.51%, respectively. This latent group has previously been described in studies which applied LCA.[11 13 37 38] Likewise, the other two identified classes are similar to those reported in a systematic review based on 14 studies of multimorbidity patterns.[39] In this review, the most prevalent diseases in the 'cardio-metabolic' group were

diabetes, hypertension, heart diseases, hyperlipidaemia and obesity; and in the second group conditions such as mental disorders, thyroid disease, neurological disease, pain, asthma or chronic lung diseases, musculoskeletal disorders, obesity and gastro-oesophageal reflux disease were included. Despite the fact that we included a smaller number of diseases, we found analogous patterns. In our study, 8.93% of the younger group (50–64 years) and 27.22% of the older were classified into the 'cardio-metabolic' class, including individuals with higher prevalence of diabetes, hypertension, myocardial infarction or angina, and stroke. This clustering of diseases is similar to the metabolic syndrome, which has metabolically related cardiovascular risk factors and greater risk of stroke and diabetes.[40] Lastly, the least prevalent group was the 'respiratory-mental-articular' class, consisting of greater prevalence of joint disorders, asthma, chronic lung diseases and depression. Association between depression and arthritis has commonly been reported, with socioeconomic and disease factors reported as being involved in its association, as well as systemic inflammation mechanisms.[3] Nevertheless, the links between depression and chronic lung diseases, and chronic lung diseases and arthritis, despite having been studied, remain unclear.[40 41]

Analogous latent multimorbidity classes have been found among both age groups. Despite this, certain aspects should be pointed out. As expected, the proportion of participants classified into the 'healthy' class was greater in participants aged 50–64 years (87.16%) compared with those aged +65 years (67.51%), illustrating higher multimorbidity in elderly individuals. The distribution of CCs was also less clear in the older subsample.

For example, both joint disorders and angina–myocardial infarction were similarly present in the 'cardio-metabolic' and 'respiratory-mental-articular' categories, whereas in the younger participants (50–64 years) subsample we observed a more differentiated profile of those CCs that cluster into one latent class. For example, respiratory-related diseases (asthma, chronic lung diseases) are highly presented in the 'respiratory-mental-articular' class, while very infrequent among middle-aged people classified into the 'cardio-metabolic' group. It is worth mentioning that although depression is frequently observed among participants classified into the 'respiratory-mental-articular' class, it is not infrequent among people within the 'cardio-vascular' class. This may be due to the relationship between mental and physical disorders, which has frequently been reported, suggesting a bidirectional association between them.[42] On the one hand, medical conditions could be accompanied by a high symptom burden, leading to depression, and, on the other, depression could be a risk factor for medical conditions, since depressive symptoms could increase the incidence of behaviours, such as smoking, alcohol intake, poor diet or physical inactivity, which are risk factors for NCDs.[3 42]

One important implication of our findings is the relatively high proportion of people aged 50–64 years with multimorbidity. Thus, preventive and intervention programmes are also needed for this population to mitigate the multimorbidity burden.

Our results show that these multimorbidity patterns are qualitatively different, but only when compared with the 'healthy' class in terms of sociodemographic and economic characteristics, lifestyles and health status variables. As has been reported in the literature, being older, woman, widowed, with a lower level of education and lower socioeconomic status are related to an increased risk of multimorbidity.[3] In addition, those individuals with multiple CCs were more likely to have limitations in ADL, especially those classified into the 'respiratory-mental-articular' group, similar to what was found in another study of multimorbidity.[37] Physical activity seems to be a protective factor for being classified into the 'respiratory-mental-articular' class, whereas smokers were more likely to be classified into the 'respiratory-mental-articular' class, but not the 'cardio-metabolic' class. This is inconsistent with the literature, since cigarette smoking is considered a major cause of cardiovascular diseases (CVDs). However, smoking is probably the most complex and least understood risk factor for CVDs.[43]

One interesting finding is the association between cognition outcomes and multimorbidity in both age subsamples. Better performance in verbal memory and fluency was related to less risk of being classified into the multimorbidity groups, with similar results among latent classes. Impaired cognition has been associated with conditions such as arthritis,[44] depression[45] and respiratory diseases,[46] cardiovascular conditions, diabetes,[47] hypertension[48] and coronary heart diseases.[49]

Concerning the regional distribution of multimorbidity, Russia accounted for the highest burden as opposed to Africa, China and India. The 'cardio-metabolic' class is especially common in this country, with a prevalence of 18.82% in the younger and 48.34% in the older subsample. Prevalence of CVDs, such as hypertension, myocardial infarction or angina, and stroke, was also higher in Russia. This high proportion could be related to the high rate of alcohol consumption and rapid societal changes experienced in this country, which might account for increased risk of circulatory diseases.[50 51] Followed by Russia, European regions showed higher rates of multimorbidity. NCDs such as hypertension, joint disorders, respiratory diseases and depression were highly prevalent, especially in England and Southern Europe, where the 'respiratory-mental-articular' class was highly prevalent in both age subsamples. The relationship between mood disorders such as depression and joint disorders has been previously reported in other studies, though the underlying cause remains unclear.[36 37 39] Notwithstanding, previous studies suggested that the emotional burden of joint disorders may contribute to the onset of psychiatric disorders.[36 52]

LMICs such as Africa, China and India showed lower rates of multimorbidity compared with Russia and other HICs. However, there was a wide variation in terms of some diseases, such as respiratory diseases and depression. Asthma and chronic lung diseases were highly prevalent in India and China, influenced by factors such as increasing smoking rates, air pollution and occupational lung diseases in these countries.[53] As reported in previous studies,[54] depression was remarkably prevalent in India, whereas the lowest prevalence was observed in China. This is in line with previous epidemiological studies on the prevalence of depression in Chinese older people, suggesting differences in diagnostic criteria that make depression less diagnosed; somatic symptoms are more prevalent in this population instead of sadness, and lack of interest and energy. Moreover, stigma and prejudice in Chinese population might also contribute to under-reporting depressive symptoms.[55 56] Furthermore, the variation found across regions in terms of depression prevalence could be due to cultural differences in expressions or expectations of mood disorders or mental health.[57]

The highest burden of multimorbidity in HICs could be explained by an increased level of development in the HICs. Notwithstanding, LMICs are experiencing a change in lifestyle and environmental exposures, which contributes, as in HICs, to multimorbidity. Thus, the increased burden of NCDs, in addition to the existing burden of infectious diseases such as HIV/AIDS, worsens multimorbidity management.[3] Moreover, the differences found in the regional distribution of multimorbidity might be linked to different stages of development of their health systems, since there are differences between LMICs and HICs in terms of opportunities and barriers to improving the organisation, integration and delivery of multimorbidity care.[3]

## Strengths and limitations

A major strength of this study is the use of a large, harmonised, multiregional database. Research on multimorbidity has typically been hampered by several factors, such as the exclusion of patients with multimorbidity from participation, targeting of research mostly on elderly individuals and a shortage of studies focusing on LMICs. The ATHLOS study allowed us to compare two age groups (50–64 and 65 years or older) as well as disease prevalence and clusters of conditions in regions with differing incomes in a very large, diverse population-based study of middle-aged and older adults.

Some limitations should be considered when interpreting our findings. First, the presence or absence of the NCDs was based on self-reported measures, and thus might be affected by measurement errors or lack of accuracy. Nevertheless, some authors sustain self-reported diagnostics as a well-established method for the measurement of multimorbidity in population-based studies.[58] Second, participants with an incipient neurodegenerative disease may have been included in our analytical sample. However, we excluded those participants who completed a proxy interview due to cognitive problems, such as neurodegenerative diseases, which could affect the reliability of the data. Nonetheless, participants with an incipient neurodegenerative disease may have been included in our analytical sample because of the lack of strong diagnostic criteria for dementia in the included studies. Third, we could only focus on those diseases that were common across studies. Conditions such as obesity, cancer, kidney disease and neurological illness were not evaluated. This might have led to a smaller number of latent classes or to different patterns of multimorbidity. Fourth, when performing LCA, the three-class solution was forced. In order to determine whether the latent classes were equivalent, invariance analysis should have been performed.[59] Nevertheless, this solution was forced as we aimed to do comparisons among age subsamples and regions in terms of disease prevalence as well as protective and risk factors. Finally, the use of multiple imputations could carry some bias. Despite this, the use of multiple imputation procedures is widely advocated when missing data occur in one or more covariates in a regression model and under an MAR assumption.[60 61]

The results of this study suggest that NCDs cluster together in non-random associations across several regions worldwide. The three qualitatively distinct entities are also linked to several sociodemographic and economic characteristics, lifestyles and health status variables. A deeper understanding of the interactions across regions and the studied variables is needed. Knowledge regarding broad patterns of conditions may contribute to the creation and implementation of guidelines that consider clusters of conditions instead of single diseases, since multimorbidity has become an unavoidable reality. Future efforts should focus on the underlying mechanisms of these clusters as well as their stability over time using longitudinal data. Moreover, cohort and age effects should be explored as might influence the likelihood of reporting some diagnosis and hence result in different multimorbidity patterns.

**Author affiliations**
[1]Research, Innovation and Teaching Unit, Parc Sanitari Sant Joan de Déu, Sant Boi de Llobregat, Spain
[2]Centro de Investigación Biomédica en Red de Salud Mental, CIBERSAM, Madrid, Spain
[3]Department of Medicine, Universitat de Barcelona, Barcelona, Spain
[4]Center of Epidemiological Studies of HIV/AIDS and STI of Catalonia (CEEISCAT), Health Department, Generalitat de Catalunya, Badalona, Spain
[5]Health Service and Population Research Department, Institute of Psychiatry, Psychology and Neuroscience, King's College London, London, United Kingdom
[6]Serra Húnter fellow. Department of Statistics and Operations Research, Polytechnic University of Catalonia, Barcelona, Spain

**Acknowledgements** The English Longitudinal Study of Ageing (ELSA): ELSA is supported by the US National Institute of Aging, the National Centre for Social Research, the University College London (UCL) and the Institute for Fiscal Studies. The authors gratefully acknowledge the UK Data Service and UCL who provided data for this paper. The Study on Global Ageing and Adult Health (SAGE): The SAGE study is funded by the US National Institute on Aging and has received financial support through Interagency Agreements (OGHA 04034785; YA1323-08-CN-0020; Y1-AG-1005–01) and Grants (R01-AG034479; IR21-AG034263-0182). The authors gratefully acknowledge the WHO who provided data for this paper. The Survey of Health, Ageing and Retirement in Europe (SHARE): The SHARE study is funded by the European Commission through FP5 (QLK6-CT-2001-00360), FP6 (SHARE-I3: RII-CT-2006-0 62 193, COMPARE: CIT5-CT-2005-0 28 857, SHARELIFE: CIT4-CT-2006-0 28 812) and FP7 (SHARE-PREP: N°211909, SHARE-LEAP: N°227822, SHARE M4: N°261982). Additional funding from the German Ministry of Education and Research, the Max Planck Society for the Advancement of Science, the US National Institute on Aging (U01_AG09740-13S2, P01_AG005842, P01_AG08291, P30_AG12815, R21_AG025169, Y1-AG-4553-01, IAG_BSR06-11, OGHA_04-064, HHSN271201300071C) and from various national funding sources is gratefully acknowledged (see www.share-project.org).

**Contributors** IB-M participated in the database management, drafted the paper, carried out the statistical analyses and worked on the interpretation of data; AS-N participated in the study design, database management, statistical support and critical revision of the paper; LE-C participated in the interpretation of data and critical revision of the paper; HN participated in critical revision of the paper; AMP participated in the study design and critical revision of the paper; DF participated in the study design, database management, statistical support and critical revision of the paper; JMH participated in the study design, acquisition of data, interpretation of data and critical revision of the paper; BO participated in the acquisition of data, study design, database management and critical revision of the paper. All authors gave final approval of the version to be published and agreed to be accountable for all aspects of the work in ensuring that questions related to the accuracy or integrity of any part of the work are appropriately investigated and resolved.

**Funding** This work was supported by the 5-year Ageing Trajectories of Health: Longitudinal Opportunities and Synergies (ATHLOS) project and the Centro de InvestigaciónBiomédica en Red de Salud Mental (CIBERSAM). The ATHLOS project has received funding from the European Union's Horizon 2020 research and innovation programme under grant agreement No 635 316. DF's work has been supported by grant RTI2018-100927-J-I00 RetosInvestigación from Ministerio de Ciencia e Innovación (MCI), by Marsden grant E2987-3648 (Royal Society of New Zealand), and by grant 2017 SGR 622 (GRBIO) from the Departament d'Economia i Coneixement de la Generalitat de Catalunya (Spain). This work, grant number RTI2018-100927-J-I00, is supported by the Ministerio de Ciencia e Innovación (MCI, Spain), by the AgenciaEstatal de Investigación (AEI, Spain) and by the European Regional Development Fund FEDER (FEDER, UE). BO's work is supported by the PERIS programme 2016–2020 'Ajuts per a la Incorporació de CientíficsiTecnòlegs' (grant number SLT006/17/00066), with the support of the Health Department from the Generalitat de Catalunya.

**Competing interests** None declared.

**Patient consent for publication** Not required.

**Ethics approval** The study protocol was approved by the Committee on the Ethics of Clinical Research, CEIC FundacióSant Joan de Déu (Protocol No: PIC-22–15). All

data were anonymised and EHR confidentially was respected in accordance with national and international law.

**Provenance and peer review** Not commissioned; externally peer reviewed.

**Data availability statement** Data are available upon reasonable request. Data may be obtained from a third party and are not publicly available. The original studies data are available on their respective websites: the Study on Global Ageing and Adult Health—SAGE (https://www.who.int/healthinfo/sage/en/), the English Longitudinal Study of Ageing—ELSA (https://www.elsa-project.ac.uk/), and the Survey of Health, Ageing and Retirement in Europe—SHARE (http://www.share-project.org/home0.html). R codes for harmonising the included variables, as well as the STATA codes for the performed analysis are available on https://github.com/athlosproject/athlos-project.github.io.

**Open access** This is an open access article distributed in accordance with the Creative Commons Attribution 4.0 Unported (CC BY 4.0) license, which permits others to copy, redistribute, remix, transform and build upon this work for any purpose, provided the original work is properly cited, a link to the licence is given, and indication of whether changes were made. See: https://creativecommons.org/licenses/by/4.0/.

**ORCID iDs**
Ivet Bayes-Marin http://orcid.org/0000-0002-3816-5244
Albert Sanchez-Niubo http://orcid.org/0000-0003-0309-181X
Laia Egea-Cortés http://orcid.org/0000-0003-4002-8254
Hai Nguyen http://orcid.org/0000-0003-2171-1955
Matthew Prina http://orcid.org/0000-0001-6698-3263
Daniel Fernández http://orcid.org/0000-0003-0012-2094
Josep Maria Haro http://orcid.org/0000-0002-3984-277X
Beatriz Olaya http://orcid.org/0000-0003-2046-3929

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
