## [Reviewer comments · BMJ Open]

ARTICLE DETAILS

TITLE (PROVISIONAL)	Multimorbidity patterns in low-middle and high income regions: A multi-region latent class analysis using ATHLOS harmonized cohorts.
AUTHORS	Bayes-Marin, Iveta; Sanchez-Niubo, Albert; Egea-Cortés, Laia; Nguyen, Hai; Prina, A. Matthew; Fernández, Daniel; Haro, Josep; Olaya, Beatriz

VERSION 1 – REVIEW

REVIEWER	Serhiy Dekhtyar Aging Research Center. Karolinska Institutet, Sweden
REVIEW RETURNED	11-Nov-2019

GENERAL COMMENTS	Authors present a latent class analysis of multimorbidity clusters from harmonized data across high income and low income countries. They then relate latent multimorbidity clusters with a series of covariates to describe potential determinants of multimorbidity. Understanding the composition of multimorbidity represents is an exciting area of inquiry, especially from a regional comparison perspective. The paper is generally well-written, and augmented. My comments are as follows: 1. Although the authors have a regional focus in mind, they derive the latent classes of multimorbidity on a pooled study population. I was wondering about the possibility of conducting LCA separately within the regional samples. This way, new patterns of disease clustering could potentially become apparent.2. I was wondering about the relevance of important auxiliary variables for the computation of latent classes. For instance, age and gender could have important influence over how the latent classes are assigned in the first place. I realize that authors explore the effects of these two variables in relation to class membership later, but could they consider using these relevant covariates already in the computation of disease clusters?3. In some examples of earlier studies on multimorbidity, hypertension has been excluded from the multimorbidity count, as it was viewed more of a risk factor for the development of overt cardiovascular disease, rather than a disease of its own. Would the same classes emerge if hypertension were not included as one of the index diseases?4. I was puzzled by the regional differences in depression prevalence. Southern and Western Europe had 5-6 times greater prevalence of depression than Russia. How would authors explain these differences? Are these real differences, artefacts of diagnostic procedures, differences in mental health bias or propensities to seek help for mental health issues?5. Related is the issue of cut-offs for depression that were used. Could the authors specify those explicitly?
--

	6. Although the data originates from studies using longitudinal designs, the analyses presented here is cross-sectional. This has to be made more apparent throughout the study, especially with respect to covariate analyses, where temporal ordering between exposures and multimorbidity clusters cannot be established. 7. A related issue has to do with the timing of collection of the diagnoses in the different source databases. Period differences can have a significant bearing on the likelihood of reporting some diagnosis as opposed to the others. What waves (and years) of the different data sources were used in the analyses? Can authors reflect/adjust for the different time periods of outcome assessment? 8. Given that multimorbidity is defined based on self-reported diagnoses of medical conditions, how did the authors approach recollection issues in individuals with dementia or cognitive impairment? 9. The analytical strategy in Tables 3 and 4 was somewhat unclear to me. It is unclear to me why certain variables are treated as covariates (and their parameters are excluded from the “adjusted” analyses, whereas others are considered as exposures of interest and their estimates are shown [eg: marital status vs. feeling of loneliness]). Also, I am not sure I see the use for fully unadjusted analyses without controls for age or sex. Could authors change unadjusted to minimally-adjusted instead, while also presenting the parameters on the full range of covariates in the “adjusted” section of the table? 10. Since the authors are interested in the regional differences, one way of examining them would be via interactions between geographical regions and covariates of interest (eg. Russia x wealth quintile). Did authors examine interactions?
--	---

REVIEWER	Beibei Xu Peking University, China
REVIEW RETURNED	11-Dec-2019

GENERAL COMMENTS	This innovative study combined data from different cohorts to investigate the difference of multimorbidity patterns and their associated factors in low- and middle-income countries and high income countries among older adults. However, there were a few questions to be mentioned. 1. For the whole manuscript, abbreviations should be correctly used that they should be declared the first time it was presented in Abstract and in Main Text. 2. Please be consistent in the use of sex or gender, also the use of younger group or youngest group. Abstract 1. The abbreviations like HICs in the Conclusion should be announced the first time it was used. Background 1. The abbreviation of LMICs should be announced the first time it showed. 2. It would be beneficial to cite previous studies on multimorbidity patterns in different LMICs and HICs. Methods
--

	1. It is plausible that Mexico was excluded due to the high percentage of missing value. However, at what percentage of other countries missing in the variables of interest? 2. It seems that the assessment of depression divergent in these cohorts so the cut-off value for the scores were different. I believe that you have used the proper cut-off value based on previous evidence, but I think it is better for you to cite the reference. 3. Though ADL might not be the key variable in this study, it should also be clearly presented that the items used in each cohort. Actually, you declared six items on Page 6 Line 28-30 but coded "yes" for the persons reported at least one difficulty in any of the five items. It is confusing that what the five items refer to. 4. The abbreviation of confidence intervals should be "CI" and 95%CI should be the abbreviation of 95% confidence interval. The same for all tables. 5. How did you use interpretability and judgment when determining the number of latent classes? 6. It is troubling to assume that missing data were assumed as missing at random. Although multiple imputation is appropriate for data missing completely at random (MCAR), it may not be an option for health-related diagnoses. The authors should describe the pattern of missingness in the dataset. 7. Once the abbreviation of MAR was declared, it can be used directly with no need for a secondary announcement. 8. Multinomial logistic regression models were used in this study. However, it seems that the data was a longitudinal data (and you also used RRR in the results). In that case, it was improper to use logistic regression if the data was a repeated measure. Maybe GEE models or Mixed-effect models will be more suitable. 9. It may cause some trouble that you adjusted for income and education level in the same model without testing the collinear. There may be a strong correlation between these two covariates, especially in HICs. Results 1. The exclusion criteria should be presented in Methods rather than in the Results. 2. The description of the prevalence of NCDs should be better organized. See the description of diabetes at Paragraph 3 Page 8. 3. It may be more appropriate that asthma and chronic lung disease being combined as one morbidity as asthma is a chronic lung disease. Combining these conditions would be more in line with the combining of several different joint disorders as one condition as you have done. 5. "Below 0.70" but not ".70". 4. Please be consistent in the use of the term 3-class or three-class.
--	--

	5. You should summarize the results in Table 3 and Table 4 but not only repeated the results. Discussion 1. For “Medical disorders”, did you mean physical disorders or somatic disorder? 2. The difference of multimorbidity among regions may be partially owing to the difference of medical/health care system, which should be discussed. 3. It may be better to add some discussion about implications of the findings.
--	--

VERSION 1 – AUTHOR RESPONSE

Reviewer #1 (Serhiy Dekhtyar, Aging Research Center. Karolinska Institutet, Sweden)

COMMENT 5: “Although the authors have a regional focus in mind, they derive the latent classes of multimorbidity on a pooled study population. I was wondering about the possibility of conducting LCA separately within the regional samples. This way, new patterns of disease clustering could potentially become apparent.”

RESPONSE: That is a very interesting remark. Actually, we conducted a separate Latent Class Analysis (LCA) for each regional sample at the beginning of our research. The results obtained were the same disease clustering structure we got using the pooled study population instead. According to several sources (see for instance Nylund-Gibson K, Choi AY. (2018) or the LCA plugin STATA manual (Lanza et al., 2018)), it is a proper measure to include into the model as a cluster a region-based effect, which specifies that individuals are nested within clusters. We stated this rationale with this sentence (page 9, lines 9-11): “Region was used as a cluster when conducting LCA to accurately describe disease proportions, indicating that the subjects were not independent random draws, but rather were nested within clusters”.

References:

Nylund-Gibson K, Choi AY. Ten frequently asked questions about latent class analysis. *Transl Issues Psychol Sci* 2018;4:440–61. doi:10.1037/tps0000176

Lanza ST, Dziak JJ, Huang L WA, Collins L. *LCA Stata plugin users ’ guide (version 1.2.1)*. Penn State: : University Park: The Methodology Center 2018.

COMMENT 6: “I was wondering about the relevance of important auxiliary variables for the computation of latent classes. For instance, age and gender could have important influence over how the latent classes are assigned in the first place. I realize that authors explore the effects of these two variables in relation to class membership later, but could they consider using these relevant covariates already in the computation of disease clusters?”

RESPONSE: This is a relevant comment with regard to methodology since the inclusion of covariates into an LCA model is a very active area of methodological research. We are aware

3

that it is possible to include covariates when conducting an LCA. However, we chose not to include them in the LCA stage because the inclusion of an auxiliary variable can and may unintentionally have influence over the final structure of the latent class, both in relative class size and type. This feature is observed by Nylund-Gibson K, Choi AY. (2018). For that reason, as our main goal was to identify groups only based on disease variables, we dismissed the inclusion of covariates such as sex and age as we did not want to determine the class structures by them. To clarify this point, we included the following sentence (page 9, lines 6-9): “Latent Class Analysis (LCA) was conducted stratified by age (50-64, +65). Eight NCDs (diabetes, hypertension, asthma, chronic lung disease, joint disorders, angina-myocardial infarction, stroke, and depression) were used as observed indicators without using covariates

since we aimed to identify latent classes only based on disease variables". As the reviewer observed, the effect of those variables was later assessed when performing the multinomial regression analyses.

References:

Nylund-Gibson K, Choi AY. Ten frequently asked questions about latent class analysis. *Transl Issues Psychol Sci* 2018;4:440–61. doi:10.1037/tps0000176

COMMENT 7: "In some examples of earlier studies on multimorbidity, hypertension has been excluded from the multimorbidity count, as it was viewed more of a risk factor for the development of overt cardiovascular disease, rather than a disease of its own. Would the same classes emerge if hypertension were not included as one of the index diseases?"

RESPONSE: Thank you for the question. As the reviewer says, in some previous studies hypertension was excluded from the multimorbidity count because it could be considered as a risk factor for other conditions. We follow the associative multimorbidity framework (Van den Akker, M., Buntinx, F., & Knottnerus, J. A. (1996) which is considered more informative in terms of understanding disease patterns from the perspective of nonrandom association of health problems (Prados-Torres, 2014). Additionally, other studies (Yen et al. (2014), Baldo et al. (2018), Larsen et al. (2017), Guisado-Clavero et al. (2018), Olaya et al. (2017), Garin et al. (2014)) did include hypertension in their count and we believe that including hypertension in our study would allow comparability with these studies.

References:

Van Den Akker M, Buntinx F, Knottnerus JA. Comorbidity or multimorbidity: What's in a name? A review of literature. *Eur. J. Gen. Pract.* 1996;2:65–70. doi:10.3109/13814789609162146

Prados-Torres A, Calderón-Larrañaga A, Hanco-Saavedra J, et al. Multimorbidity patterns: A systematic review. *J Clin Epidemiol* 2014;67:254–66. doi:10.1016/j.jclinepi.2013.09.021

Yen L, Valderas JM, McRae IS, et al. Multimorbidity and comorbidity of chronic diseases among the senior Australians: prevalence and patterns. *PLoS One* 2014;9:e83783

Baldo V, Corti MC, Boccuzzo G, et al. Multimorbidity patterns in high-need, high-cost elderly patients. *PLoS One* 2018;13:e0208875. doi:10.1371/journal.pone.0208875

Larsen FB, Pedersen MH, Friis K, et al. A Latent class analysis of multimorbidity and the relationship to socio-demographic factors and health-related quality of life. A national population-based study of 162,283 Danish Adults. *PLoS One* 2017;12:1–17. doi:10.1371/journal.pone.0169426

4

Guisado-Clavero M, Roso-Llorach A, López-Jimenez T, et al. Multimorbidity patterns in the elderly: A prospective cohort study with cluster analysis. *BMC Geriatr* 2018;18:1–11. doi:10.1186/s12877-018-0705-7

Olaya B, Moneta MV, Caballero FF, et al. Latent class analysis of multimorbidity patterns and associated outcomes in Spanish older adults: A prospective cohort study. *BMC Geriatr* 2017;17. doi:10.1186/s12877-017-0586-1

Garin N, Olaya B, Perales J, et al. Multimorbidity patterns in a national representative sample of the Spanish adult population. *PLoS One* 2014;9:e84794. doi:10.1371/journal.pone.0084794

COMMENT 8: "I was puzzled by the regional differences in depression prevalence. Southern and Western Europe had 5-6 times greater prevalence of depression than Russia. How would authors explain these differences? Are these real differences, artefacts of diagnostic procedures, differences in mental health bias or propensities to seek help for mental health issues?"

RESPONSE: Again, thank you for this interesting question. We acknowledge that there are important differences in the depression prevalence rates across regions/countries. First, we checked whether the prevalence rates found in our study were similar to that reported in the original study and found that they are the same. As the reviewer mentions, one potential reason could be found in different diagnosis/assessment procedures in these studies. Three studies were included in this study: ELSA, SAGE and SHARE. Each study used a different

assessment tool, which is described in depth in comment number 9.

Russia took part in the SAGE study using similar diagnostic criteria as the regions involved in the SHARE study, despite using different tools. Moreover, Castro-Costa et al. (2008) analysed the depression prevalence across European regions obtaining similar results as we did. In their study, they suggested differences could be due to culturally determined differences in norms or expectations or expressions of mood and mental health. We agree with this explanation, as we considered the cultural effect when explaining the observed differences in depression in the Chinese population. Thus, we included this sentence (page 21, lines 8-9): “Furthermore, the variation found across regions in terms of depression prevalence could be due to cultural differences in expressions or expectations of mood disorders or mental health”.

References:

Castro-Costa E, Dewey M, Stewart R, et al. Ascertaining late-life depressive symptoms in Europe: An evaluation of the survey version of the EURO-D scale in 10 nations. The SHARE project. *Int J Methods Psychiatr Res* 2008;17:12–29. doi:10.1002/mpr.236

COMMENT 9: “Related is the issue of cut-offs for depression that were used. Could the authors specify those explicitly?”

RESPONSE: We thank the reviewer for the opportunity to clarify this point. Depression was assessed using different tools in each study and we used the proposed cut-off for each study population. As part of the ATHLOS project, depression was harmonized across studies to allow comparability despite using different tools. Rationale and methodology of ATHLOS project can be found here: <https://academic.oup.com/ije/article/48/4/1052/5477844> (Sanchez-Niubo et al. (2019)).

Briefly, these are the cut-off points used to calculate and harmonize depression:

5

- In the SAGE study, nine possible symptoms of depression were included in addition to a duration criteria (2 weeks). If at least five symptoms and the duration criteria were met, depression was coded as “yes”. The algorithm was derived from The World Mental Health (WMH) Survey Initiative Version of the World Health Organization (WHO) Composite International Diagnostic Interview (CIDI), Kessler & Üstün (2004).
- ELSA used the 8-item version of the Center for Epidemiologic Studies Depression Scale (CES-D). To determine depressive symptoms, each self-respondent was asked nine questions with response options of ‘yes’ or ‘no’. Note that in this version, there were dichotomised items instead of Likert-scale items. For each respondent, the total number of “yes” responses to questions 1, 2, 3, 5, 7, 8, and the “no” responses to questions 4 and 6 (reversed items) were summed to arrive at a total depressive symptom score ranging from 0 to 8. We classified those who reported four or more depressive symptoms as having significant depressive symptoms, a cut-off that has been found to produce comparable results to the 16-symptom cut-off for the wellvalidated 20-item CES-D scale (Zivin et al. 2010).
- SHARE study used the EURO-D, constituted by 12 items scored by summing item scores for individual symptoms that are coded as 0 and 1 when they are “not present” and “present”, respectively. In this case we used the dichotomised variable created by SHARE researchers, using 4 or more as a proper cut-off score (Prince et al. 1999).

References:

Sanchez-Niubo A, Egea-Cortés L, Olaya B et al. Cohort profile: the ageing trajectories of health - longitudinal opportunities and synergies (ATHLOS) project. *Int J Epidemiol* 2019;0:1–11. doi:org/10.1093/ije/dyz077

Ronald C. Kessler TBÜ. The World Mental Health (WMH) survey initiative version of the World Health Organization (WHO) Composite Diagnostic Interview (CIDI). *Int J Methods Psychiatr Res* 2004;13:93–121. doi:org/10.1002/mpr.168

Zivin K, Llewellyn DJ, Lang IA, et al. Depression among older adults in the United States and England. *Am J Geriatr Psychiatry* 2010;18:1036–44. doi:10.1097/JGP.0b013e3181dba6d2

Prince MJ, Beekman ATF, Deeg DJH, et al. Depression symptoms in late life assessed during the EURO-D scale. Effect of age, gender and marital status in 14 European centres. *Br J Psychiatry* 1999;174:339–45. doi:10.1192/bjp.174.4.339

COMMENT 10: “Although the data originates from studies using longitudinal designs, the analyses presented here is cross-sectional. This has to be made more apparent throughout the study, especially with respect to covariate analyses, where temporal ordering between exposures and multimorbidity clusters cannot be established.”

RESPONSE: We apologize for the lack of clarity in our original submission. We have included some sentences in order to clarify that despite being longitudinal studies, we performed cross-sectional analyses. “All the analyses were performed using data from the baseline” (page 9, line 2); “Multinomial logistic regression models were used to assess the association of each multimorbidity class with several outcomes adjusted for gender, sex, age, marital status, education level, wealth, and the region at baseline” (page 9, lines 24-26).

COMMENT 11: “A related issue has to do with the timing of collection of the diagnoses in the different source databases. Period differences can have a significant bearing on the

6 likelihood of reporting some diagnosis as opposed to the others. What waves (and years) of the different data sources were used in the analyses? Can authors reflect/adjust for the different time periods of outcome assessment?”

RESPONSE: Thank you so much for the suggestion. It might be very interesting to compare different time periods. We used the first wave of the three studies: 1998 (ELSA), 2004 (SHARE) and 2007 (SAGE). Since the difference across time collection events is not extremely wide, we did not consider including an interaction between age and age of birth when conducting the multinomial regression analyses. However, we acknowledge this as a limitation and point out as a future research objective (page 22, lines 21-22): “Future efforts should focus on the underlying mechanisms of these clusters as well as their stability over time using longitudinal data. Moreover, cohort and age effects should be explored as might influence the likelihood of reporting some diagnosis and hence result in different multimorbidity patterns.”

COMMENT 12: “Given that multimorbidity is defined based on self-reported diagnoses of medical conditions, how did the authors approach recollection issues in individuals with dementia or cognitive impairment?”

RESPONSE: Our analytical sample is based on individuals who completed the interview by themselves through direct interviews. Those participants with dementia or suspected cognitive impairment participated via a proxy respondent. When performing the analyses, we excluded those participants with proxy interviews because this kind of interviews is usually shorter than the direct ones. Thus, individuals with dementia or cognitive impairment are thought to be excluded from the analyses. The following sentence has been added in the “Study design and data extraction” section (page 7, lines 5-6): “We excluded from the analyses those participants who participated via proxy due to cognitive problems or severe physical limitations”.

COMMENT 13: “The analytical strategy in Tables 3 and 4 was somewhat unclear to me. It is unclear to me why certain variables are treated as covariates (and their parameters are excluded from the “adjusted” analyses, whereas others are considered as exposures of interest and their estimates are shown [eg: marital status vs. feeling of loneliness]). Also, I am not sure I see the use for fully unadjusted analyses without controls for age or sex. Could authors change unadjusted to minimally-adjusted instead, while also presenting the parameters on the full range of covariates in the “adjusted” section of the table?”

RESPONSE: Following both reviewers’ comments, we decided to reformulate Tables 3 and 4. Now we only show the adjusted results integrating both age subsamples and presenting the unadjusted results in a Supplementary file (Additional file 2.docx).

COMMENT 14: “Since the authors are interested in the regional differences, one way of examining them would be via interactions between geographical regions and covariates of

interest (eg. Russia x wealth quintile). Did authors examine interactions?"

RESPONSE: Thanks you so much for this suggestion. Of course, including interactions would be an interesting direction to explore. However, that would be difficult to address in our case due to a large number of regions and the two age subsamples. Moreover, we used imputed data doing it even more cumbersome. We could consider examining interactions in future research when analysing a lower number of regions. Thus, we included the following section in the discussion (page 22, lines 16-17):

7

"The three qualitatively distinct entities are also linked to several sociodemographic and economic characteristics, lifestyles, and health status variables. A deeper understanding of the interactions across regions and the studied variables is needed."

Reviewer #2 (Beibei Xu, Peking University, China)

COMMENT 15: "For the whole manuscript, abbreviations should be correctly used that they should be declared the first time it was presented in Abstract and in Main Text."

RESPONSE: We reviewed the whole document and we amended it.

COMMENT 16: "Please be consistent in the use of sex or gender, also the use of younger group or youngest group."

RESPONSE: We homogenised those terms and changed gender for sex because is more appropriate since those studies registered the biological sex. We also changed "youngest" for "younger".

COMMENT 17: "Abstract - The abbreviations like HICs in the Conclusion should be announced the first time it was used."

RESPONSE: You are right. We corrected it.

COMMENT 18: "Background - The abbreviation of LMICs should be announced the first time it showed."

RESPONSE: We reviewed the whole document in order to avoid this kind of inaccuracies.

COMMENT 19: "Background - It would be beneficial to cite previous studies on multimorbidity patterns in different LMICs and HICs."

RESPONSE: We included a summary of studies in different LMICs and HICs (page 5, lines 12-28): "Most studies on the prevalence of multimorbidity in older people come from HICs, while data from middle-aged adults and low- and middle- income countries (LMICs) are much more limited [5–8]. LMICs are experiencing an increase in life expectancy that, together with changes in lifestyle and environment exposures, are triggering changes in their disease burden profile [3,9]. Few studies have compared patterns of multimorbidity between HICs and LMICs. Afshar et al. [10] used population-based chronic disease data from the World Health Survey to compare multimorbidity prevalence across 27 LMICs and one HIC, and used gross domestic product (GDP) to study inter-country socioeconomic differences. They found high multimorbidity prevalence in all the studied countries, and a positive but non-linear relationship between country GDP and multimorbidity prevalence, suggesting the influence of other factors, such as lifestyles, social conditions, and differences among health systems. Four latent classes were identified in a cross-sectional sample of Australian seniors aged 50 years and over, using a self-reported diagnosis of eleven conditions, including cancer and Parkinson's disease [11]. Another study, focusing on complex health care needs of Italian elderly people, found five clusters using 15 diseases [12]. A study conducted in a sample of 162,283 people from a survey of the Danish population identified seven latent classes considering 15 chronic diseases and seven age groups, ranging from 16 to 104 years [13]. These differences could be

8

explained in light of variations in collection methods, data sources, populations, diseases included, and the analysis performed [11,14,15]."

COMMENT 20: "Methods - It is plausible that Mexico was excluded due to the high percentage of missing value. However, at what percentage of other countries missing in the variables of interest?"

RESPONSE: With the aim of making it visible we prepared a Supplementary material (See Additional file 1.docx: Table S1-10) with the percentage of missingness for each region in the variables of interest.

COMMENT 21: “Methods - It seems that the assessment of depression divergent in these cohorts so the cut-off value for the scores were different. I believe that you have used the proper cut-off value based on previous evidence, but I think it is better for you to cite the reference.”

RESPONSE: We thank the reviewer for the opportunity to clarify this point. Depression was assessed using different tools in each study and we used the proposed cut-off for each study population. Due to the limited space, we included the corresponding reference of each depression instrument in the manuscript.

COMMENT 22: “Methods - Though ADL might not be the key variable in this study, it should also be clearly presented that the items used in each cohort. Actually, you declared six items on Page 6 Line 28-30 but coded “yes” for the persons reported at least one difficulty in any of the five items. It is confusing that what the five items refer to.”

RESPONSE: Thank you so much for spotting this. It was a mistake. The ADL variable is made up with six items. We corrected it and clarified that those six items were common across studies.

COMMENT 23: Methods- The abbreviation of confidence intervals should be “CI” and 95%CI should be the abbreviation of 95% confidence interval. The same for all tables.

RESPONSE: Thank you. We corrected it.

COMMENT 24: Methods - How did you use interpretability and judgment when determining the number of latent classes?

RESPONSE: Based on previous literature, when deciding on the number of latent classes the best option is to jointly consider statistical fit indices, substantive interpretability, and classification diagnostics, which allow us to determine how well the classes are classifying and differentiating among the individuals considered (Nylund-Gibson, 2018; Masyn, 2013). Therefore, we considered classification diagnoses and similar patterns of multimorbidity reported in previous studies to determine the clinical sense of a latent class.

References:

Nylund-Gibson K, Choi AY. Ten frequently asked questions about latent class analysis. *Transl Issues Psychol Sci* 2018;4:440–61. doi:10.1037/tps0000176

9

Masyn KE. Latent Class Analysis and Finite Mixture Modeling. Published Online First: 21 March 2013. doi:10.1093/OXFORDHB/9780199934898.013.0025

COMMENT 25: “Methods - It is troubling to assume that missing data were assumed as missing at random. Although multiple imputation is appropriate for data missing completely at random (MCAR), it may not be an option for health-related diagnoses. The authors should describe the pattern of missingness in the dataset.”

RESPONSE: We performed multiple imputation (MI) assuming missing at random (MAR) based on the directions of several authors, since MI procedure builds on the MAR assumption, but the method can handle also MCAR and MNAR. According to the bibliography consulted, when including enough variables predictive of missing values in the imputation model, MI lead us to correct biases (Pedersen et al. (2017), Rubin et al (1987), Sterne et al. (2009)). We included several variables in our MI model and performed graphical diagnostics, which improves our confidence in this procedure.

As you suggested, we have included a Supplementary material (Additional file 1.docx) which describes the pattern of missingness in the dataset.

References:

Pedersen AB, Mikkelsen EM, Cronin-Fenton D, et al. Missing data and multiple imputation in clinical epidemiological research. *Clin Epidemiol* 2017;9:157–66. doi:10.2147/CLEP.S129785

Rubin DB. Multiple Imputation for nonresponse in surveys. United States of America: : John

Wiley & Sons 1987. doi:10.2307/3172772

Sterne JAC, White IR, Carlin JB, et al. Multiple imputation for missing data in epidemiological and clinical research: Potential and pitfalls. *BMJ* 2009;339:157–60. doi:10.1136/bmj.b2393

COMMENT 26: “Methods - Once the abbreviation of MAR was declared, it can be used directly with no need for a secondary announcement.”

RESPONSE: The reviewer is right. We changed it.

COMMENT 27: “Methods - Multinomial logistic regression models were used in this study. However, it seems that the data was a longitudinal data (and you also used RRR in the results). In that case, it was improper to use logistic regression if the data was a repeated measure. Maybe GEE models or Mixed-effect models will be more suitable.”

RESPONSE: We are grateful to the reviewer’s comment. Despite using studies with a longitudinal design, our aim was answered by focusing only on cross-sectional data (baseline waves). Thus, multinomial logistic regression models would appear as suitable. We included some sentences throughout the manuscript to clarify that we performed cross-sectional analyses using baseline data.

COMMENT 28: “Methods - It may cause some trouble that you adjusted for income and education level in the same model without testing the collinear. There may be a strong correlation between these two covariates, especially in HICs.”

RESPONSE: Thanks for pointing this out. We agree that there might be a strong correlation between income and education. Thus, we tested the collinearity using Mantel Haenszel chi

10 square test (MH) and saw that there is a significant association between both covariates. Then, we performed a polychromic correlation to check the magnitude of this association and we found that this association was small (0.3075964 (50-64 years old subsample) 0.3169671 (65+ old subsample). Therefore, we believe that there would not be collinearity problems in our model. We included this sentence in the manuscript (page 9, line 26): “Due to potential collinearity between income and education, we checked the significance and magnitude of the correlation between both variables. The association was small, and thus both covariates were included as separate variables in the models”.

COMMENT 29: “Results - The exclusion criteria should be presented in Methods rather than in the Results.”

RESPONSE: We moved the exclusion criteria into the methods section.

COMMENT 30: “Results -The description of the prevalence of NCDs should be better organized. See the description of diabetes at Paragraph 3 Page 8.”

RESPONSE: Thanks. This paragraph has been re-written to make it clearer.

COMMENT 31: “Results -It may be more appropriate that asthma and chronic lung disease being combined as one morbidity as asthma is a chronic lung disease. Combining these conditions would be more in line with the combining of several different joint disorders as one condition as you have done.”

RESPONSE: The “joint disorders” variable was a result of the harmonization process of the ATHLOS study since the studies reported different disorders, such as rheumatoid arthritis, arthritis, osteoarthritis.

As for the combination of asthma and chronic lung disease, we decided to respect the list of conditions from the original studies because chronic lung disease includes chronic bronchitis or emphysema, which are more severe conditions. Moreover, previous studies on multimorbidity patterns used asthma and chronic lung disease as separated observed indicators and we intended to allow comparability.

COMMENT 32: Results - “Below 0.70” but not “ .70”.

RESPONSE: Thank you for pointing out these issues. It was a typing error, which has been corrected in the revised version of the manuscript.

COMMENT 33: “Results - Please be consistent in the use of the term 3-class or three-class.”

RESPONSE: It is consisted now. We used “three-class”.

COMMENT 34: “Results - You should summarize the results in Table 3 and Table 4 but not only repeated the results.”

RESPONSE: We reviewed again the document and realised that there was some duplication. Thus, we unified Table 3 and 4 presenting only the adjusted results and summarizing them in the text. The unadjusted results have been moved into a supplementary material (Additional file 2.docx: Table S1).

11

COMMENT 35: “Discussion - For “Medical disorders”, did you mean physical disorders or somatic disorder?”

RESPONSE: We meant physical disorders. For sure, using “physical disorders” is better than “medical disorders”, so we changed it for the former. Thank you very much for your comment.

COMMENT 36: Discussion - The difference of multimorbidity among regions may be partially owing to the difference of medical/health care system, which should be discussed.

RESPONSE: Thanks for your suggestion. We agree that difference of medical care systems might explain the differences found, so we mention the possible impact of health systems across regions on multimorbidity (page 21, lines 14-17): “Moreover, the differences found in the regional distribution of multimorbidity might be linked to different stages of development of their health systems, since there are differences between LMICs and HICs in terms of opportunities and barriers to improving the organization, integration, and delivery of multimorbidity care [3]. ”.

COMMENT 37: Discussion - It may be better to add some discussion about implications of the findings.

RESPONSE: We thank the reviewer for this comment, which allows us to add such an important point of the manuscript. We have included a paragraph regarding implications of the findings as well as suggestion for future research (page 22, lines 15-24): “The results of this study suggest that NCDs cluster together in non-random associations across several regions worldwide. The three qualitatively distinct entities are also linked to several sociodemographic and economic characteristics, lifestyles, and health status variables. A deeper understanding of the interactions across regions and the studied variables is needed. Knowledge regarding broad patterns of conditions may contribute to the creation and implementation of guidelines that consider clusters of conditions instead of single diseases, since multimorbidity has become an unavoidable reality. Future efforts should focus on the underlying mechanisms of these clusters as well as their stability over time using longitudinal data. Moreover, cohort and age effects should be explored as might influence the likelihood of reporting some diagnosis and hence result in different multimorbidity patterns.”

VERSION 2 – REVIEW

REVIEWER	Serhiy Dekhtyar Karolinska Institutet, Sweden
REVIEW RETURNED	04-May-2020

GENERAL COMMENTS	Authors adequately addressed my comments. Only thing, I would like that diagnostic criteria for dementia, used in the different studies are included as part of the supplementary analysis (see my comment # 9.
---

REVIEWER	Beibei Xu Peking University Health Science Center
REVIEW RETURNED	05-May-2020

GENERAL COMMENTS	The manuscript is overall well-written. One suggestion is that more details can be added to better clarify the multinomial logistic regression analyses in the methodology part. What were the outcomes and how were these outcomes defined in the models? Were they continuous or categorical? Regarding to Table 3, were smoking and physical activity the outcomes in the models? Smoking and physical activity were usually considered as lifestyle confounders for examining association between multimorbidity and memories or other health outcomes. Why not including smoking and physical activity as confounders for the other models?
--

VERSION 2 – AUTHOR RESPONSE

Reviewer #1 (Serhiy Dekhtyar, Aging Research Center, Karolinska Institutet, Sweden)

COMMENT: “Authors adequately addressed my comments. Only thing, I would like that diagnostic criteria for dementia, used in the different studies are included as part of the supplementary analysis (see my comment #9).”

RESPONSE: We thank the reviewer for the positive general appraisal of our study. Those individuals in the three studies with suspected cognitive impairment, physical or mental ill health participated via a proxy respondent. We excluded those participants with proxy interviews because their interviews were shorter than those performed with the direct participants. In that manner, individuals with cognitive impairment or dementia were likely excluded from the analysis.

As for dementia diagnosis criteria, some differences among studies should be pointed out here. Firstly, dementia was defined using two indicators (Hackett et al., 2018) in the ELSA study. The first one was self-reported physician-diagnosed dementia. The other one was obtained as an average score of 3.5 or greater from the Informant Questionnaire on Cognitive Decline (IQCODE) (Jorm, 1994). The IQCODE is a screening tool that comprises 16 questions, which are asked to a proxy informant, about a set of cognitive abilities compared them to two years ago. Secondly, some filtering questions with regards to memory were asked in the SAGE study at the beginning of the interview to check whether the respondent was able to complete the interview. If he/she was unable to do so, the interviewer provided the questionnaire to the proxy respondent instead and the respondent was excluded from the sample (Peltzer & Phaswana-Mafuya, 2012). Then, the IQCODE was used with proxy informants to determine dementia diagnosis (Study of Global Ageing and Adult Health (SAGE), 2006). Finally, information about dementia status was only available at waves 2 to 5 in the SHARE study, but not at baseline (Sterniczuk et al., 2015).

Due to both this heterogeneity and lack of information regarding dementia diagnosis, we considered that including it as a supplementary analysis is not the best option. Nevertheless, we think that this topic fits better in the limitation section, as follows (Page 23, lines 15-20):

“Secondly, participants with an incipient neurodegenerative disease may have been included in our analytical sample. However, we excluded those participants who completed a proxy interview due to cognitive problems, such as neurodegenerative diseases, which could affect the reliability of the data. Nonetheless, participants with an incipient neurodegenerative disease may have been included in our analytical sample because of the lack of strong diagnostic criteria for dementia in the included studies.”

References:

Hackett RA, Davies-Kershaw H, Cadar D, et al. Walking Speed, Cognitive Function, and Dementia Risk in the English Longitudinal Study of Ageing. *J Am Geriatr Soc* 2018;66:1670–5. doi:10.1111/jgs.15312

Jorm AF. A Short Form of the Informant Questionnaire on Cognitive Decline in the Elderly (Iqcode): Development and Cross-Validation. *Psychol Med* 1994;24:145–53. doi:10.1017/S003329170002691X

Peltzer K, Phaswana-Mafuya N. Cognitive functioning and associated factors in older adults in South Africa. *South African J Psychiatry* 2012;18:157–63. doi:10.7196/SAJP.368
Study of Global Ageing and Adult Health (SAGE) SURVEY MANUAL. 2006.

Sterniczuk R, Theou O, Rusak B, et al. Cognitive test performance in relation to health and function in 12 European countries: The SHARE study. *Can Geriatr J* 2015;18:144–51. doi:10.5770/cgj.18.154

Reviewer #2 (Beibei Xu, Peking University, China)

COMMENT: “The manuscript is overall well-written. One suggestion is that more details can be added to better clarify the multinomial logistic regression analyses in the methodology part. What were the outcomes and how were these outcomes defined in the models? Were they continuous or categorical? Regarding to Table 3, were smoking and physical activity the outcomes in the models? Smoking and physical activity were usually considered as lifestyle confounders for examining association between multimorbidity and memories or other health outcomes. Why not including smoking and physical activity as confounders for the other models?”

RESPONSE: We appreciated the reviewer’s comment. We noticed that there was a mistake when we mentioned “outcomes” instead of “variables: “Multinomial logistic regression models were used to assess the association of each multimorbidity class with several outcomes”, which might drive the reader towards misleading conclusions. In our case, the outcome was the membership class which is a variable defined as a three-level categorical variable. Thus, we rephrased the following paragraph (Page 8, lines 25-26, page 9, and lines 1-3) as follows:

“Adjusted multinomial logistic regression models were used to assess the association between the outcome (multimorbidity classes, with the “Healthy” class as the reference category) and several variables: loneliness, ever smoked, physical activity, limitations in ADL, self-rated health, immediate recall, delayed recall and verbal fluency. The model was additionally adjusted for sex, age, marital status, education level, wealth, and the region at baseline”

Regarding smoking and physical activity variables, we absolutely agree with the reviewer about the huge impact that those two variables can possibly have in a multimorbidity analysis. Therefore, they were also included as independent variables in our model. We chose those variables which are commonly included as confounders in other studies on multimorbidity as we also aimed to allow comparability with previous research work (Larsen et al. (2017) Garin et al. (2014), Olaya et al. (2017), Park et al. (2019)). We included sex, age, current marital status, education level, quintiles of household wealth, and the region as confounders. Additionally, we decided to include those independent variables one by one within the model with the aim of both testing parsimonious models and assessing independently several risk factors of multimorbidity.

References:

Larsen FB, Pedersen MH, Friis K, et al. A Latent class analysis of multimorbidity and the relationship

to socio-demographic factors and health-related quality of life. A national population-based study of 162,283 Danish Adults. PLoS One 2017;12:1–17. doi:10.1371/journal.pone.0169426

Garin N, Olaya B, Perales J, et al. Multimorbidity patterns in a national representative sample of the Spanish adult population. PLoS One 2014;9:e84794. doi:10.1371/journal.pone.0084794

Olaya B, Moneta MV, Caballero FF, et al. Latent class analysis of multimorbidity patterns and associated outcomes in Spanish older adults: A prospective cohort study. BMC Geriatr 2017;17. doi:10.1186/s12877-017-0586-1

Park B, Lee HA, Park H. Use of latent class analysis to identify multimorbidity patterns and associated factors in Korean adults aged 50 years and older. PLoS One 2019;14:e0216259. doi:10.1371/journal.pone.0216259

VERSION 3 – REVIEW

REVIEWER	Serhiy Dekhtyar Aging Research Center, Karolinska Institutet. Stockholm, Sweden
REVIEW RETURNED	24-May-2020
GENERAL COMMENTS	Authors have adequately addressed my comments.